# ROUTING NETWORKS: ADAPTIVE SELECTION OF NON-LINEAR FUNCTIONS FOR MULTI-TASK LEARNING

**Clemens Rosenbaum**
College of Information and Computer Sciences
University of Massachusetts Amherst
140 Governors Dr., Amherst, MA 01003
cgbr@cs.umass.edu

**Tim Klinger & Matthew Riemer**
IBM Research AI
1101 Kitchawan Rd, Yorktown Heights, NY 10598
{tklinger,mdriemer}@us.ibm.com

## ABSTRACT

Multi-task learning (MTL) with neural networks leverages commonalities in tasks to improve performance, but often suffers from task interference which reduces the benefits of transfer. To address this issue we introduce the routing network paradigm, a novel neural network and training algorithm. A routing network is a kind of self-organizing neural network consisting of two components: a *router* and a set of one or more *function blocks*. A function block may be any neural network – for example a fully-connected or a convolutional layer. Given an input the router makes a routing decision, choosing a function block to apply and passing the output back to the router recursively, terminating when a fixed recursion depth is reached. In this way the routing network dynamically composes different function blocks for each input. We employ a collaborative multi-agent reinforcement learning (MARL) approach to jointly train the router and function blocks. We evaluate our model against cross-stitch networks and shared-layer baselines on multi-task settings of the MNIST, mini-imagenet, and CIFAR-100 datasets. Our experiments demonstrate a significant improvement in accuracy, with sharper convergence. In addition, routing networks have nearly constant per-task training cost while cross-stitch networks scale linearly with the number of tasks. On CIFAR-100 (20 tasks) we obtain cross-stitch performance levels with an 85% reduction in training time.

## 1 INTRODUCTION

Multi-task learning (MTL) is a paradigm in which multiple tasks must be learned simultaneously. Tasks are typically separate prediction problems, each with their own data distribution. In an early formulation of the problem, (Caruana, 1997) describes the goal of MTL as improving generalization performance by "leveraging the domain-specific information contained in the training signals of related tasks." This means a model must leverage commonalities in the tasks (positive transfer) while minimizing interference (negative transfer). In this paper we propose a new architecture for MTL problems called a *routing network*, which consists of two trainable components: a *router* and a set of *function block*s. Given an input, the router selects a function block from the set, applies it to the input, and passes the result back to the router, recursively up to a fixed recursion depth. If the router needs fewer iterations then it can decide to take a PASS action which leaves the current state unchanged. Intuitively, the architecture allows the network to dynamically self-organize in response to the input, sharing function blocks for different tasks when positive transfer is possible, and using separate blocks to prevent negative transfer.

The architecture is very general allowing many possible router implementations. For example, the router can condition its decision on both the current activation and a task label or just one or the other. It can also condition on the depth (number of router invocations), filtering the function module choices to allow layering. In addition, it can condition its decision for one instance on what was historically decided for other instances, to encourage re-use of existing functions for improved compression. The function blocks may be simple fully-connected neural network layers or whole

networks as long as the dimensionality of each function block allows composition with the previous function block choice. They needn't even be the same type of layer. Any neural network or part of a network can be "routed" by adding its layers to the set of function blocks, making the architecture applicable to a wide range of problems. Because the routers make a sequence of hard decisions, which are not differentiable, we use reinforcement learning (RL) to train them. We discuss the training algorithm in Section 3.1, but one way we have modeled this as an RL problem is to create a separate RL agent for each task (assuming task labels are available in the dataset). Each such task agent learns its own policy for routing instances of that task through the function blocks.

To evaluate we have created a "routed" version of the convnet used in (Ravi & Larochelle, 2017) and use three image classification datasets adapted for MTL learning: a multi-task MNIST dataset that we created, a Mini-imagenet data split as introduced in (Vinyals et al., 2016), and CIFAR-100 (Krizhevsky, 2009), where each of the 20 label superclasses are treated as different tasks.[1] We conduct extensive experiments comparing against cross-stitch networks (Misra et al., 2016) and the popular strategy of joint training with layer sharing as described in (Caruana, 1997). Our results indicate a significant improvement in accuracy over these strong baselines with a speedup in convergence and often orders of magnitude improvement in training time over cross-stitch networks.

## 2 RELATED WORK

Work on multi-task deep learning (Caruana, 1997) traditionally includes significant hand design of neural network architectures, attempting to find the right mix of task-specific and shared parameters. For example, many architectures share low-level features like those learned in shallow layers of deep convolutional networks or word embeddings across tasks and add task-specific architectures in later layers. By contrast, in routing networks, we learn a fully dynamic, compositional model which can adjust its structure differently for each task.

Routing networks share a common goal with techniques for automated selective transfer learning using attention (Rajendran et al., 2017) and learning gating mechanisms between representations (Stollenga et al., 2014), (Misra et al., 2016), (Ruder et al., 2017). In the latter two papers, experiments are performed on just 2 tasks at a time. We consider up to 20 tasks in our experiments and compare directly to (Misra et al., 2016).

Our work is also related to mixtures of experts architectures (Jacobs et al., 1991), (Jordan & Jacobs, 1994) as well as their modern attention based (Riemer et al., 2016) and sparse (Shazeer et al., 2017) variants. The gating network in a typical mixtures of experts model takes in the input and chooses an appropriate weighting for the output of each expert network. This is generally implemented as a soft mixture decision as opposed to a hard routing decision, allowing the choice to be differentiable. Although the sparse and layer-wise variant presented in (Shazeer et al., 2017) does save some computational burden, the proposed end-to-end differentiable model is only an approximation and doesn't model important effects such as exploration vs. exploitation tradeoffs, despite their impact on the system. Mixtures of experts have recently been considered in the transfer learning setting (Aljundi et al., 2016), however, the decision process is modelled by an autoencoder-reconstruction-error-based heuristic and is not scaled to a large number of tasks.

In the use of dynamic representations, our work is also related to single task and multi-task models that learn to generate weights for an optimal neural network (Ha et al., 2016), (Ravi & Larochelle, 2017), (Munkhdalai & Yu, 2017). While these models are very powerful, they have trouble scaling to deep models with a large number of parameters (Wichrowska et al., 2017) without tricks to simplify the formulation. In contrast, we demonstrate that routing networks can be applied to create dynamic network architectures for architectures like convnets by routing some of their layers.

Our work extends an emerging line of recent research focused on automated architecture search. In this work, the goal is to reduce the burden on the practitioner by automatically learning black box algorithms that search for optimal architectures and hyperparameters. These include techniques based on reinforcement learning (Zoph & Le, 2017), (Baker et al., 2017), evolutionary algorithms (Miikkulainen et al., 2017), approximate random simulations (Brock et al., 2017), and adaptive growth (Cortes et al., 2016). To the best of our knowledge we are the first to apply this idea to multi-task learning. Our technique can learn to construct a very general class of architectures without the

---

[1] All dataset splits and the code will be released with the publication of this paper.

need for human intervention to manually choose which parameters will be shared and which will be kept task-specific.

Also related to our work is the literature on minimizing computation cost for single-task problems by conditional routing. These include decisions trained with REINFORCE (Denoyer & Gallinari, 2014), (Bengio et al., 2015), (Hamrick et al., 2017), Q Learning (Liu & Deng, 2017), and actor-critic methods (McGill & Perona, 2017). Our approach differs however in the introduction of several novel elements. Specifically, our work explores the multi-task learning setting, it uses a multi-agent reinforcement learning training algorithm, and it is structured as a recursive decision process.

There is a large body of related work which focuses on continual learning, in which tasks are presented to the network one at a time, potentially over a long period of time. One interesting recent paper in this setting, which also uses the notion of routes ("paths"), but uses evolutionary algorithms instead of RL is Fernando et al. (2017).

While a routing network is a novel artificial neural network formulation, the high-level idea of task specific "routing" as a cognitive function is well founded in biological studies and theories of the human brain (Gurney et al., 2001), (Buschman & Miller, 2010), (Stocco et al., 2010).

## 3  ROUTING NETWORKS

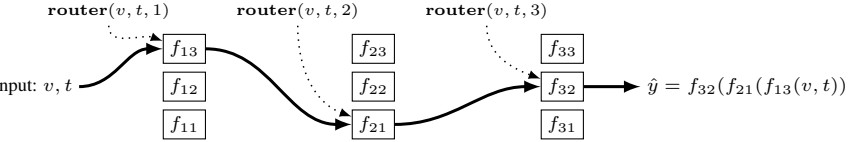

Figure 1: Routing (forward) Example

A routing network consists of two components: a *router* and a set of *function block*s, each of which can be any neural network layer. The router is a function which selects from among the function blocks given some input. *Routing* is the process of iteratively applying the router to select a sequence of function blocks to be composed and applied to the input vector. This process is illustrated in Figure 1. The input to the routing network is an instance to be classified $(v, t), v \in \mathbb{R}^d$ is a representation vector of dimension $d$ and $t$ is an integer task identifier. The router is given $v, t$ and a depth (=1), the depth of the recursion, and selects from among a set of function block choices available at depth 1, $\{f_{13}, f_{12}, f_{11}\}$, picking $f_{13}$ which is indicated with a dashed line. $f_{13}$ is applied to the input $(v, t)$ to produce an output activation. The router again chooses a function block from those available at depth 2 (if the function blocks are of different dimensions then the router is constrained to select dimensionally matched blocks to apply) and so on. Finally the router chooses a function block from the last (classification) layer function block set and produces the classification $\hat{y}$.

Algorithm 1 gives the routing procedure in detail. The algorithm takes as input a vector $v$, task label $t$ and maximum recursion depth $n$. It iterates $n$ times choosing a function block on each iteration and applying it to produce an output representation vector. A special PASS action (see Appendix Section 7.2 for details) just skips to the next iteration. Some experiments don't require a task label and in that case we just pass a dummy value. For simplicity we assume the algorithm has access to the router function and function blocks and don't include them explicitly in the input. The router decision function **router** : $\mathbb{R}^d \times \mathbb{Z}^+ \times \mathbb{Z}^+ \to \{1, 2, \ldots, k, PASS\}$ (for $d$ the input representation dimension and $k$ the number of function blocks) maps the current representation $v$, task label $t \in \mathbb{Z}^+$, and current depth $i \in \mathbb{Z}^+$ to the index of the function block to route next in the ordered set **function_block**.

---

**Algorithm 1:** Routing Algorithm

**input** : $x, t, n$:
  $x \in \mathbb{R}^d$, $d$ the representation dim;
  $t$ integer task id;
  $n$ max depth

**output:** $v$ - the vector result of applying the composition of the selected functions to the input $x$

1  $v \leftarrow x$
2  **for** $i$ *in* $1...n$ **do**
3  $\quad a \leftarrow$ **router**$(x, t, i)$
4  $\quad$ **if** $a \neq PASS$ **then**
5  $\quad \quad x \leftarrow$ **function_block**$_a(x)$
6  **return** $v$

---

If the routing network is run for $d$ invocations then we say it has *depth d*. For $N$ function blocks a routing network run to a depth $d$ can select from $N^d$ distinct trainable functions (the paths in the network). Any neural network can be represented as a routing network by adding copies of its layers as routing network function blocks. We can group the function blocks for each network layer and constrain the router to pick from layer 0 function blocks at depth 0, layer 1 blocks at depth 1, and so on. If the number of function blocks differs from layer to layer in the original network, then the router may accommodate this by, for example, maintaining a separate decision function for each depth.

### 3.1 ROUTER TRAINING USING RL

---

**Algorithm 2:** Router-Trainer: Training of a Routing Network.

---

**input:** A dataset $D$ of samples $(v, t, y)$, $v$ the input representation, $t$ an integer task label, $y$ a ground-truth target label

1 **for** *each sample* $s = (v, t, y) \in D$ **do**

2      Do a forward pass through the network, applying Algorithm 1 to sample $s$.
     Store a trace $T = (S, A, R, r_{final})$, where $S$ = sequence of visited states $(s_i)$; $A$ = sequence of actions taken $(a_i)$; $R$ = sequence of immediate action rewards $(r_i)$ for action $a_i$; and the final reward $r_{final}$.
     The last output as the network's prediction $\hat{y}$ and the final reward $r_{final}$ is +1 if the prediction $\hat{y}$ is correct; -1 if not.

3      Compute the loss $\mathcal{L}(\hat{y}, y)$ between prediction $\hat{y}$ and ground truth $y$ and backpropagate along the function blocks on the selected route to train their parameters.

4      Use the trace $T$ to train the router using the desired RL training algorithm.

---

We can view routing as an RL problem in the following way. The states of the MDP are the triples $(v, t, i)$ where $v \in \mathbb{R}^d$ is a representation vector (initially the input), $t$ is an integer task label for $v$, and $i$ is the depth (initially 1). The actions are function block choices (and PASS) in $\{1, \ldots k, PASS\}$ for $k$ the number of function blocks. Given a state $s = (v, t, i)$, the router makes a decision about which action to take. For the non-PASS actions, the state is then updated $s' = (v', t, i + 1)$ and the process continues. The PASS action produces the same representation vector again but increments the depth, so $s' = (v, t, i + 1)$. We train the router policy using a variety of RL algorithms and settings which we will describe in detail in the next section.

Regardless of the RL algorithm applied, the router and function blocks are trained jointly. For each instance we route the instance through the network to produce a prediction $\hat{y}$. Along the way we record a trace of the states $s_i$ and the actions $a_i$ taken as well as an immediate reward $r_i$ for action $a_i$. When the last function block is chosen, we record a final reward which depends on the prediction $\hat{y}$ and the true label $y$.

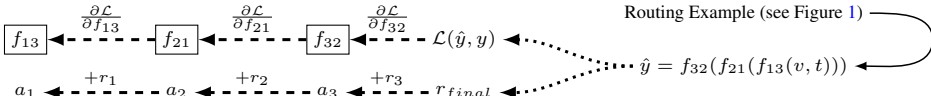

Figure 2: Training (backward) Example

We train the selected function blocks using SGD/backprop. In the example of Figure 1 this means computing gradients for $f_{32}$, $f_{21}$ and $f_{13}$. We then use the computed trace to train the router using an RL algorithm. The high-level procedure is summarized in Algorithm 2 and illustrated in Figure 2. To keep the presentation uncluttered we assume the RL training algorithm has access to the router function, function blocks, loss function, and any specific hyper-parameters such as discount rate needed for the training and don't include them explicitly in the input.

### 3.1.1 REWARD DESIGN

A routing network uses two kinds of rewards: immediate action rewards $r_i$ given in response to an action $a_i$ and a final reward $r_{final}$, given at the end of the routing. The final reward is a function

of the network's performance. For the classification problems focused on in this paper, we set it to +1 if the prediction was correct ($\hat{y} = y$), and $-1$ otherwise. For other domains, such as regression domains, the negative loss ($-\mathcal{L}(\hat{y}, y)$) could be used.

We experimented with an immediate reward that encourages the router to use fewer function blocks when possible. Since the number of function blocks per-layer needed to maximize performance is not known ahead of time (we just take it to be the same as the number of tasks), we wanted to see whether we could achieve comparable accuracy while reducing the number of function blocks ever chosen by the router, allowing us to reduce the size of the network after training. We experimented with two such rewards, multiplied by a hyper-parameter $\rho \in [0, 1]$: the average *number of times* that block was chosen by the router historically and the average historical *probability* of the router choosing that block. We found no significant difference between the two approaches and use the average probability in our experiments. We evaluated the effect of $\rho$ on final performance and report the results in Figure 12 in the appendix. We see there that generally $\rho = 0.0$ (no collaboration reward) or a small value works best and that there is relatively little sensitivity to the choice in this range.

### 3.1.2 RL Algorithms

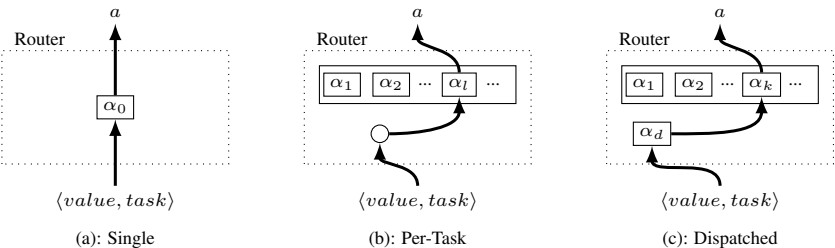

(a): Single        (b): Per-Task        (c): Dispatched

Figure 3: Task-based routing. $\langle value, task \rangle$ is the input consisting of $value$, the partial evaluation of the previous function block (or input $x$) and the task label $task$. $\alpha_i$ is a routing agent; $\alpha_d$ is a dispatching agent.

To train the router we evaluate both single-agent and multi-agent RL strategies. Figure 3 shows three variations which we consider. In Figure 3(a) there is just a single agent which makes the routing decision. This is be trained using either policy-gradient (PG) or Q-Learning experiments. Figure 3(b) shows a multi-agent approach. Here there are a fixed number of agents and a hard rule which assigns the input instance to a an agent responsible for routing it. In our experiments we create one agent per task and use the input task label as an index to the agent responsible for routing that instance. Figure 3(c) shows a multi-agent approach in which there is an additional agent, denoted $\alpha_d$ and called a *dispatching agent* which learns to assign the input to an agent, instead of using a fixed rule. For both of these multi-agent scenarios we additionally experiment with a MARL algorithm called Weighted Policy Learner (WPL).

We experiment with storing the policy both as a table and in form of an approximator. The tabular representation has the invocation depth as its row dimension and the function block as its column dimension with the entries containing the probability of choosing a given function block at a given depth. The approximator representation can consist of either one MLP that is passed the depth (represented in 1-hot), or a vector of $d$ MLPs, one for each decision/depth.

Both the Q-Learning and Policy Gradient algorithms are applicable with tabular and approximation function policy representations. We use REINFORCE (Williams, 1992) to train both the approximation function and tabular representations. For Q-Learning the table stores the Q-values in the entries. We use vanilla Q-Learning (Watkins, 1989) to train tabular representation and train the approximators to minimize the $\ell_2$ norm of the temporal difference error.

Implementing the router decision policy using multiple agents turns the routing problem into a stochastic game, which is a multi-agent extension of an MDP. In stochastic games multiple agents interact in the environment and the expected return for any given policy may change without any action on that agent's part. In this view incompatible agents need to compete for blocks to train, since negative transfer will make collaboration unattractive, while compatible agents can gain by

sharing function blocks. The agent's (locally) optimal policies will correspond to the game's Nash equilibrium [2].

For routing networks, the environment is non-stationary since the function blocks are being trained as well as the router policy. This makes the training considerably more difficult than in the single-agent (MDP) setting. We have experimented with single-agent policy gradient methods such as REINFORCE but find they are less well adapted to the changing environment and changes in other agent's behavior, which may degrade their performance in this setting.

One MARL algorithm specifically designed to address this problem, and which has also been shown to converge in non-stationary environments, is the weighted policy learner (WPL) algorithm (Abdallah & Lesser, 2006), shown in Algorithm 3. WPL is a PG algorithm designed to dampen oscillation and push the agents to converge more quickly. This is done by scaling the gradient of the expected return for an action $a$ according the probability of taking that action $\pi(a)$ (if the gradient is positive) or $1 - \pi(a)$ (if the gradient is negative). Intuitively, this has the effect of slowing down the learning rate when the policy is moving away from a Nash equilibrium strategy and increasing it when it approaches one. The full WPL algorithm is shown in Algorithm 3. It is assumed that the historical average return $\hat{\mathcal{R}}_i$ for each action $a_i$ is initialized to 0 before the start of training. The function simplex-projection projects the updated policy values to make it a valid probability distribution. The projection is defined as: $clip(\pi) / \sum (clip(\pi))$, where $clip(x) = \max(0, min(1, x))$. The states $S$ in the trace are not used by the WPL algorithm.

Details, including convergence proofs and more examples giving the intuition behind the algorithm, can be found in (Abdallah & Lesser, 2006). A longer explanation of the algorithm can be found in Section 7.4 in the appendix. The WPL-Update algorithm is defined only for the tabular setting. It is future work to adapt it to work with function approximators.

---

**Algorithm 3:** Weighted Policy Learner

**input** : A trace $T = (S, A, R, r_{final})$
$\quad\quad\quad$ $n$ the maximum depth;
$\quad\quad\quad$ $\hat{\mathcal{R}}$, the historical average returns (initialized to 0 at the start of training);
$\quad\quad\quad$ $\gamma$ the discount factor ; and
$\quad\quad\quad$ $\lambda_\pi$ the policy learning rate
**output:** An updated router policy $\pi$

1 **for** *each action $a_i \in A$* **do**
2 $\quad$ Compute the return:
3 $\quad$ $\mathcal{R}_i \leftarrow r_{final} + \sum_{j=i}^{n} \gamma^{j-i} r_j$
4 $\quad$ Update the average return:
5 $\quad$ $\hat{\mathcal{R}}_i \leftarrow (1 - \lambda_\pi)\hat{\mathcal{R}}_i + \lambda_\pi \mathcal{R}_i$
6 $\quad$ Compute the gradient:
7 $\quad$ $\Delta(a_i) \leftarrow \mathcal{R}_i - \hat{\mathcal{R}}_i$
8 $\quad$ Update the policy:
9 $\quad$ **if** $\Delta(a_i) < 0$ **then**
10 $\quad\quad$ $\Delta(a_i) \leftarrow \Delta(a_i)(1 - \pi(a_i))$
11 $\quad$ **else**
12 $\quad\quad$ $\Delta(a_i) \leftarrow \Delta(a_i)(\pi(a_i))$
13 $\quad$ $\pi \leftarrow$ simplex-projection$(\pi + \lambda_\pi \Delta)$

---

As we have described it, the training of the router and function blocks is performed independently after computing the loss. We have also experimented with adding the gradients from the router choices $\Delta(a_i)$ to those for the function blocks which produce their input. We found no advantage but leave a more thorough investigation for future work.

## 4 QUANTITATIVE RESULTS

We experiment with three datasets: multi-task versions of MNIST (MNIST-MTL) (Lecun et al., 1998), Mini-Imagenet (MIN-MTL) (Vinyals et al., 2016) as introduced by (Ravi & Larochelle, 2017), and CIFAR-100 (CIFAR-MTL) (Krizhevsky, 2009) where we treat the 20 superclasses as tasks. In the binary MNIST-MTL dataset, the task is to differentiate instances of a given class $c$ from non-instances. We create 10 tasks and for each we use 1k instances of the positive class $c$ and 1k each of the remaining 9 negative classes for a total of 10k instances per task during training, which we then test on 200 samples per task (2k samples in total). MIN-MTL is a smaller version of ImageNet (Deng et al., 2009) which is easier to train in reasonable time periods. For mini-ImageNet we randomly choose 50 labels and create tasks from 10 disjoint random subsets of 5 labels each chosen from these. Each label has 800 training instances and 50 testing instances – so 4k training and 250 testing instances per task. For all 10 tasks we have a total of 40k training instances. Finally,

---

[2]A Nash equilibrium is a set of policies for each agent where each agent's expected return will be lower if that agent unilaterally changes its policy

CIFAR-100 has coarse and fine labels for its instances. We follow existing work (Krizhevsky, 2009) creating one task for each of the 20 coarse labels and include 500 instances for each of the corresponding fine labels. There are 20 tasks with a total of 2.5k instances per task; 2.5k for training and 500 for testing. All results are reported on the test set and are averaged over 3 runs. The data are summarized in Table 1.

Each of these datasets has interesting characteristics which challenge the learning in different ways. CIFAR-MTL is a "natural" dataset whose tasks correspond to human categories. MIN-MTL is randomly generated so will have less task coherence. This makes positive transfer more difficult to achieve and negative transfer more of a problem. And MNIST-MTL, while simple, has the difficult property that the same instance can appear with different labels in different tasks, causing interference. For example, in the "0 vs other digits" task, "0" appears with a positive label but in the "1 vs other digits" task it appears with a negative label.

Our experiments are conducted on a convnet architecture (SimpleConvNet) which appeared recently in (Ravi & Larochelle, 2017). This model has 4 convolutional layers, each consisting of a 3x3 convolution and 32 filters, followed by batch normalization and a ReLU. The convolutional layers are followed by 3 fully connected layers, with 128 hidden units each. Our routed version of the network routes the 3 fully connected layers and for each routed layer we supply one randomly initialized function block per task in the dataset. When we use neural net approximators for the router agents they are always 2 layer MLPs with a hidden dimension of 64. A state $(v, t, i)$ is encoded for input to the approximator by concatenating $v$ with a 1-hot representation of $t$ (if used). That is, encoding(s) = concat$(v, \text{one\_hot}(t))$.

| Dataset | # Training | # Testing |
|---|---|---|
| CIFAR-MTL | 50k | 10k |
| MIN-MTL | 40k | 2.5k |
| MNIST-MTL | 100k | 2k |

Table 1: Dataset training and testing splits

We did a parameter sweep to find the best learning rate and $\rho$ value for each algorithm on each dataset. We use $\rho = 0.0$ (no collaboration reward) for CIFAR-MTL and MIN-MTL and $\rho = 0.3$ for MNIST-MTL. The learning rate is initialized to $10^{-2}$ and annealed by dividing by 10 every 20 epochs. We tried both regular SGD as well as Adam Kingma & Ba (2014), but chose SGD as it resulted in marginally better performance. The SimpleConvNet has batch normalization layers but we use no dropout.

For one experiment, we dedicate a special "PASS" action to allow the agents to skip layers during training which leaves the current state unchanged (routing-all-fc recurrent/+PASS). A detailed description of the PASS action is provided in the Appendix in Section 7.2.

All data are presented in Table 2 in the Appendix.

In the first experiment, shown in Figure 4, we compare different RL training algorithms on CIFAR-MTL. We compare five algorithms: MARL:WPL; a single agent REINFORCE learner with a separate approximation function per layer; an agent-per-task REINFORCE learner which maintains a separate approximation function for each layer; an agent-per-task Q learner with a separate approximation function per layer; and an agent-per-task Q learner with a separate table for each layer. The best performer is the WPL algorithm which outperforms the nearest competitor, tabular Q-Learning by about 4%. We can see that (1) the WPL algorithm works better than a similar vanilla PG, which has trouble learning; (2) having multiple agents works better than having a single agent; and (3) the tabular versions, which just use the task and depth to make their predictions, work better here than the approximation versions, which all use the representation vector in addition predict the next action.

The next experiment compares the best performing algorithm WPL against other routing approaches, including the already introduced REINFORCE: single agent (for which WPL is not applicable). All of these algorithms route the full-connected layers of the SimpleConvNet using the layering approach we discussed earlier. To make the next comparison clear we rename MARL:WPL to *routing-all-fc* in Figure 5 to reflect the fact that it routes all the fully connected layers of the SimpleConvNet, and rename REINFORCE: single agent to *routing-all-fc single agent*. We compare against several other approaches. One approach, *routing-all-fc-recurrent/+PASS*, has the same setup as *routing-all-fc*, but does not constrain the router to pick only from layer 0 function blocks at depth 0, etc. It is allowed to choose any function block from two of the layers (since the first two routed layers

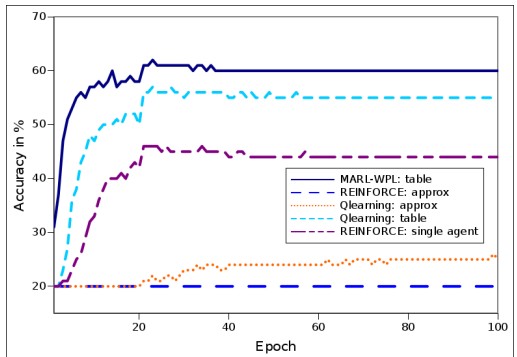
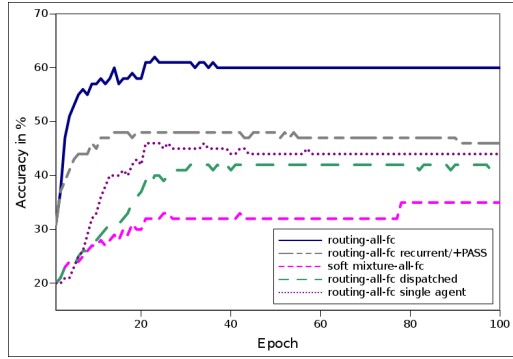

Figure 4: Influence of the RL algorithm on CIFAR-MTL. Detailed descriptions of the implementation each approach can be found in the Appendix in Section 7.3.

Figure 5: Comparison of Routing Architectures on CIFAR-MTL. Implementation details of each approach can be found in the Appendix in Section 7.3.

have identical input and output dimensions; the last is the classification layer). Another approach, *soft-mixture-fc*, is a soft version of the router architecture. This soft version uses the same function blocks as the routed version, but replaces the hard selection with a trained softmax attention (see the discussion below on cross-stitch networks for the details). We also compare against the single agent architecture shown in 3(a) called *routing-all-fc single agent* and the dispatched architecture shown in Figure 3(c) called *routing-all-fc dispatched*. Neither of these approached the performance of the per-task agents. The best performer by a large margin is *routing-all-fc*, the fully routed WPL algorithm.

We next compare *routing-all-fc* on different domains against the cross-stitch networks of Misra et al. (2016) and two challenging baselines: *task specific-1-fc* and *task specific-all-fc*, described below.

Cross-stitch networks Misra et al. (2016) are a kind of linear-combination model for multi-task learning. They maintain one model per task with a shared input layer, and "cross stitch" connection layers, which allow sharing between tasks. Instead of selecting a single function block in the next layer to route to, a cross-stitch network routes to all the function blocks simultaneously, with the input for a function block $i$ in layer $l$ given by a linear combination of the activations computed by all the function blocks of layer $l-1$. That is: $\text{input}_{li} = \sum_{j=1}^{k} w_{ij}^{l} v_{l-1,j}$, for learned weights $w_{ij}^{l}$ and layer $l-1$ activations $v_{l-1,j}$. For our experiments, we add a cross-stitch layer to each of the routed layers of SimpleConvNet. We additional compare to a similar "soft routing" version *soft-mixture-fc* in Figure 5. Soft-routing uses a softmax to normalize the weights used to combine the activations of previous layers and it shares parameters for a given layer so that $\mathbf{w_i^l} = \mathbf{w_{i'}^l}$ for all $i, i', l$.

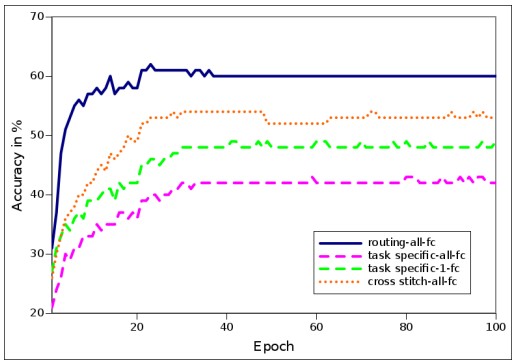
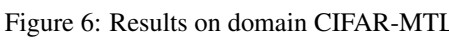

Figure 6: Results on domain CIFAR-MTL

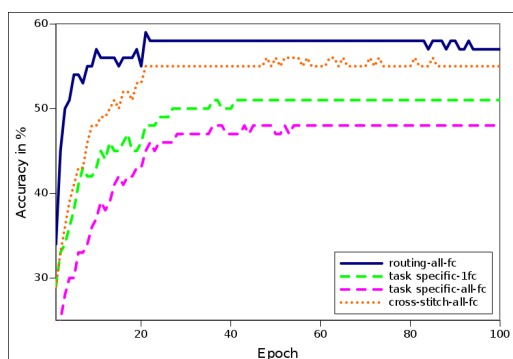

Figure 7: Results on domain MIN-MTL (mini ImageNet)

The *task-specific-1-fc* baseline has a separate last fully connected layer for each task and shares the rest of the layers for all tasks. The *task specific-all-fc* baseline has a separate set of all the fully connected layers for each task. These baseline architectures allow considerable sharing of parameters but also grant the network private parameters for each task to avoid interference. However, unlike routing networks, the choice of which parameters are shared for which tasks, and which parameters are task-private is made statically in the architecture, independent of task.

The results are shown in Figures 6, 7, and 8. In each case the routing net *routing-all-fc* performs consistently better than the cross-stitch networks and the baselines. On CIFAR-MTL, the routing net beats cross-stitch networks by 7% and the next closest baseline *task-specific-1-fc* by 11%. On MIN-MTL, the routing net beats cross-stitch networks by about 2% and the nearest baseline *task-specific-1-fc* by about 6%. We surmise that the results are better on CIFAR-MTL because the task instances have more in common whereas the MIN-MTL tasks are randomly constructed, making sharing less profitable.

On MNIST-MTL the random baseline is 90%. We experimented with several learning rates but were unable to get the cross-stitch networks to train well here. Routing nets beats the cross-stitch networks by 9% and the nearest baseline (*task-specific-all-fc*) by 3%. The soft version also had trouble learning on this dataset.

In all these experiments routing makes a significant difference over both cross-stitch networks and the baselines and we conclude that a dynamic policy which learns the function blocks to compose on a per-task basis yields better accuracy and sharper convergence than simple static sharing baselines or a soft attention approach.

In addition, router training is much faster. On CIFAR-MTL for example, training time on a stable compute cluster was reduced from roughly 38 hours to 5.6, an 85% improvement. We have conducted a set of scaling experiments to compare the training computation of routing networks and cross-stitch networks trained with 2, 3, 5, and 10 function blocks. The results are shown in the appendix in Figure 15. Routing networks consistently perform better than cross-stitch networks and the baselines across all these problems. Adding function blocks has no apparent effect on the computation involved in training routing networks on a dataset of a given size. On the other hand, cross-stitch networks has a soft routing policy that scales computation linearly with the number of function blocks. Because the soft policy backpropagates through all function blocks and the hard routing policy only backpropagates through the selected block, the hard policy can much more easily scale to many task learning scenarios that require many diverse types of functional primitives.

To explore why the multi-agent approach seems to do better than the single-agent, we manually compared their policy dynamics for several CIFAR-MTL examples. For these experiments $\rho = 0.0$ so there is no collaboration reward which might encourage less diversity in the agent choices. In the cases we examined we found that the single agent often chose just 1 or 2 function blocks at each depth, and then routed all tasks to those. We suspect that there is simply too little signal available to the agent in the early, random stages, and once a bias is established its decisions suffer from a lack of diversity.

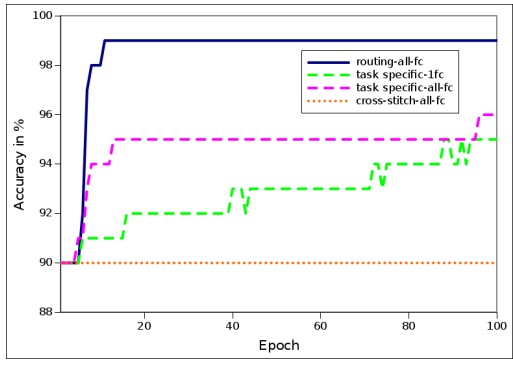

Figure 8: Results on domain MNIST-MTL

The routing network on the other hand learns a policy which, unlike the baseline static models, partitions the network quite differently for each task, and also achieves considerable diversity in its choices as can be seen in Figure 11. This figure shows the routing decisions made over the whole MNIST MTL dataset. Each task is labeled at the top and the decisions for each of the three routed layers are shown below. We believe that because the routing network has separate policies for each task, it is less sensitive to a bias for one or two function blocks and each agent learns more independently what works for its assigned task.

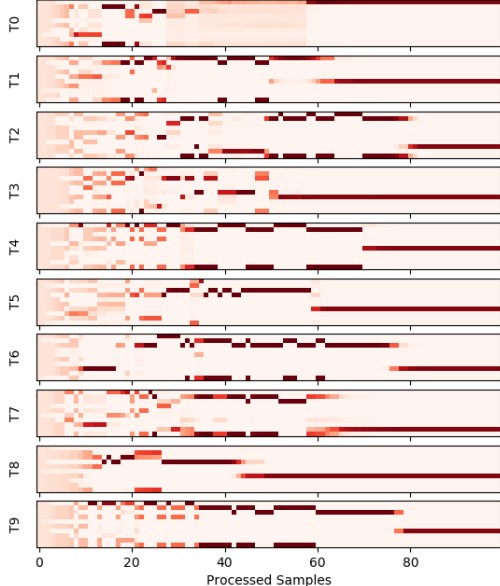

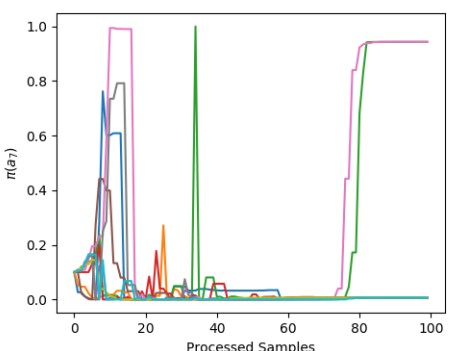

Figure 10: The Probabilities of all Agents of taking Block 7 for the first 100 samples of each task (totalling 1000 samples) of MNIST-MTL

Figure 9: The Policies of all Agents for the first function block layer for the first 100 samples of each task of MNIST-MTL

## 5    QUALITATIVE RESULTS

To better understand the agent interaction we have created several views of the policy dynamics. First, in Figure 9, we chart the policy over time for the first decision. Each rectangle labeled $T_i$ on the left represents the evolution of the agent's policy for that task. For each task, the horizontal axis is number of samples per task and the vertical axis is actions (decisions). Each vertical slice shows the probability distribution over actions after having seen that many samples of its task, with darker shades indicating higher probability. From this picture we can see that, in the beginning, all task agents have high entropy. As more samples are processed each agent develops several candidate function blocks to use for its task but eventually all agents converge to close to 100% probability for one particular block. In the language of games, the agents find a pure strategy for routing.

In the next view of the dynamics, we pick one particular function block (block 7) and plot the probability, for each agent, of choosing that block over time. The horizontal axis is time (sample) and the vertical axis is the probability of choosing block 7. Each colored curve corresponds to a different task agent. Here we can see that there is considerable oscillation over time until two agents, pink and green, emerge as the "victors" for the use of block 7 and each assign close to 100% probability for choosing it in routing their respective tasks. It is interesting to see that the eventual winners, pink and green, emerge earlier as well as strongly interested in block 7. We have noticed this pattern in the analysis of other blocks and

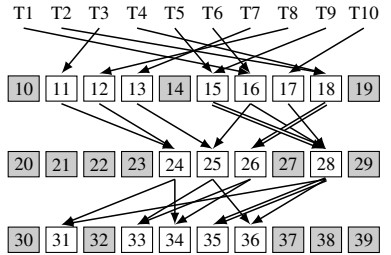

Figure 11:    An actual routing map for MNIST-MTL.

speculate that the agents who want to use the block are being pulled away from their early Nash equilibrium as other agents try to train the block away.

Finally, in Figure 11 we show a map of the routing for MNIST-MTL. Here tasks are at the top and each layer below represents one routing decision. Conventional wisdom has it that networks will benefit from sharing early, using the first layers for common representations, diverging later to accommodate differences in the tasks. This is the setup for our baselines. It is interesting to see

that this is not what the network learns on its own. Here we see that the agents have converged on a strategy which first uses 7 function blocks, then compresses to just 4, then again expands to use 5. It is not clear if this is an optimal strategy but it does certainly give improvement over the static baselines.

## 6 FUTURE WORK

We have presented a general architecture for routing and multi-agent router training algorithm which performs significantly better than cross-stitch networks and baselines and other single-agent approaches. The paradigm can easily be applied to a state-of-the-art network to allow it to learn to dynamically adjust its representations.

As described in the section on Routing Networks, the state space to be learned grows exponentially with the depth of the routing, making it challenging to scale the routing to deeper networks in their entirety. It would be interesting to try hierarchical RL techniques (Barto & Mahadevan (2003)) here.

Our most successful experiments have used the multi-agent architecture with one agent per task, trained with the Weighted Policy Learner algorithm (Algorithm 3). Currently this approach is tabular but we are investigating ways to adapt it to use neural net approximators.

We have also tried routing networks in an online setting, training over a sequence of tasks for few shot learning. To handle the iterative addition of new tasks we add a new routing agent for each and overfit it on the few shot examples while training the function modules with a very slow learning rate. Our results so far have been mixed, but this is a very useful setting and we plan to return to this problem.

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

# 7  APPENDIX

## 7.1  IMPACT OF RHO

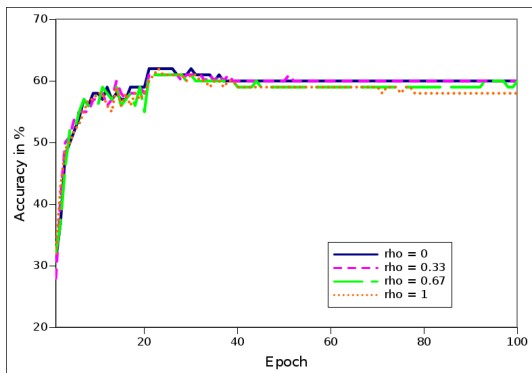

Figure 12: Influence of the "collaboration reward" $\rho$ on CIFAR-MTL. The architecture is routing-all-fc with WPL routing agents.

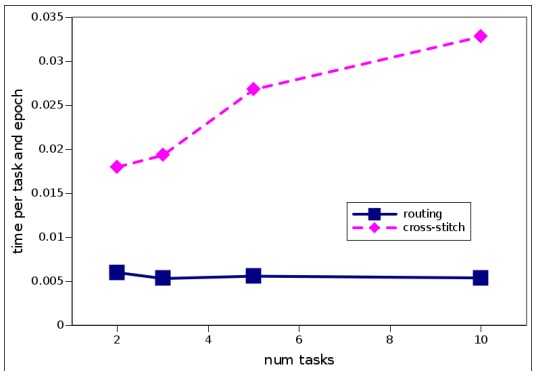

Figure 13: Comparison of per-task training cost for cross-stitch and routing networks. We add a function block per task and normalize the training time per epoch by dividing by the number of tasks to isolate the effect of adding function blocks on computation.

## 7.2  THE PASS ACTION

When routing networks, some resulting sets of function blocks can be applied repeatedly. While there might be other constraints, the prevalent one is dimensionality - input and output dimensions need to match. Applied to the SimpleConvNet architecture used throughout the paper, this means that of the fc layers - $(convolution \rightarrow 48), (48 \rightarrow 48), (48 \rightarrow \#classes)$, the middle transformation can be applied an arbitrary number of times. In this case, the routing network becomes fully recurrent and the PASS action is applicable. This allows the network to shorten the recursion depth.

## 7.3  OVERVIEW OF IMPLEMENTATIONS

We have tested 9 different implementation variants of the routing architectures. The architectures are summarized in Tables 3 and 4. The columns are:

**#Agents**    refers to how many agents are used to implement the router. In most of the experiments, each router consists of one agent per task. However, as described in 3.1, there are implementations with 1 and #tasks + 1 agents.

| | Epoch | 1 | 5 | 10 | 20 | 50 | 100 |
|---|---|---|---|---|---|---|---|
| RL (Figure 4) | REINFORCE: approx | 20 | 20 | 20 | 20 | 20 | 20 |
| | Qlearning: approx | 20 | 20 | 20 | 20 | 24 | 25 |
| | Qlearning: table | 20 | 36 | 47 | 50 | 55 | 55 |
| | MARL-WPL: table | 31 | 53 | 57 | 58 | 60 | 60 |
| arch (Figure 5) | routing-all-fc | 31 | 53 | 57 | 58 | 60 | 60 |
| | routing-all-fc recursive | 31 | 43 | 45 | 48 | 48 | 46 |
| | routing-all-fc dispatched | 20 | 23 | 28 | 37 | 42 | 41 |
| | soft mixture-all-fc | 20 | 24 | 27 | 30 | 32 | 35 |
| | routing-all-fc single agent | 20 | 23 | 33 | 42 | 44 | 44 |
| CIFAR (Figure 6) | routing-all-fc | 31 | 53 | 57 | 58 | 60 | 60 |
| | task specific-all-fc | 21 | 29 | 33 | 36 | 42 | 42 |
| | task specific-1-fc | 27 | 34 | 39 | 42 | 48 | 49 |
| | cross stitch-all-fc | 26 | 37 | 42 | 49 | 52 | 53 |
| MIN (Figure 7) | routing-all-fc | 34 | 54 | 57 | 55 | 58 | 57 |
| | task specific-all-fc | 22 | 30 | 37 | 43 | 47 | 48 |
| | task specific-1fc | 29 | 38 | 43 | 46 | 51 | 51 |
| | cross-stitch-all-fc | 29 | 41 | 48 | 53 | 56 | 55 |
| MNIST (Figure 8) | routing-all-fc | 90 | 90 | 98 | 99 | 99 | 99 |
| | task specific-all-fc | 90 | 91 | 94 | 95 | 95 | 96 |
| | task specific-1fc | 90 | 90 | 91 | 92 | 93 | 95 |
| | soft mixture-all-fc | 90 | 90 | 90 | 90 | 90 | 90 |
| | cross-stitch-all-fc | 90 | 90 | 90 | 90 | 90 | 90 |

Table 2: Numeric results (in % accuracy) for Figures 4 through 8

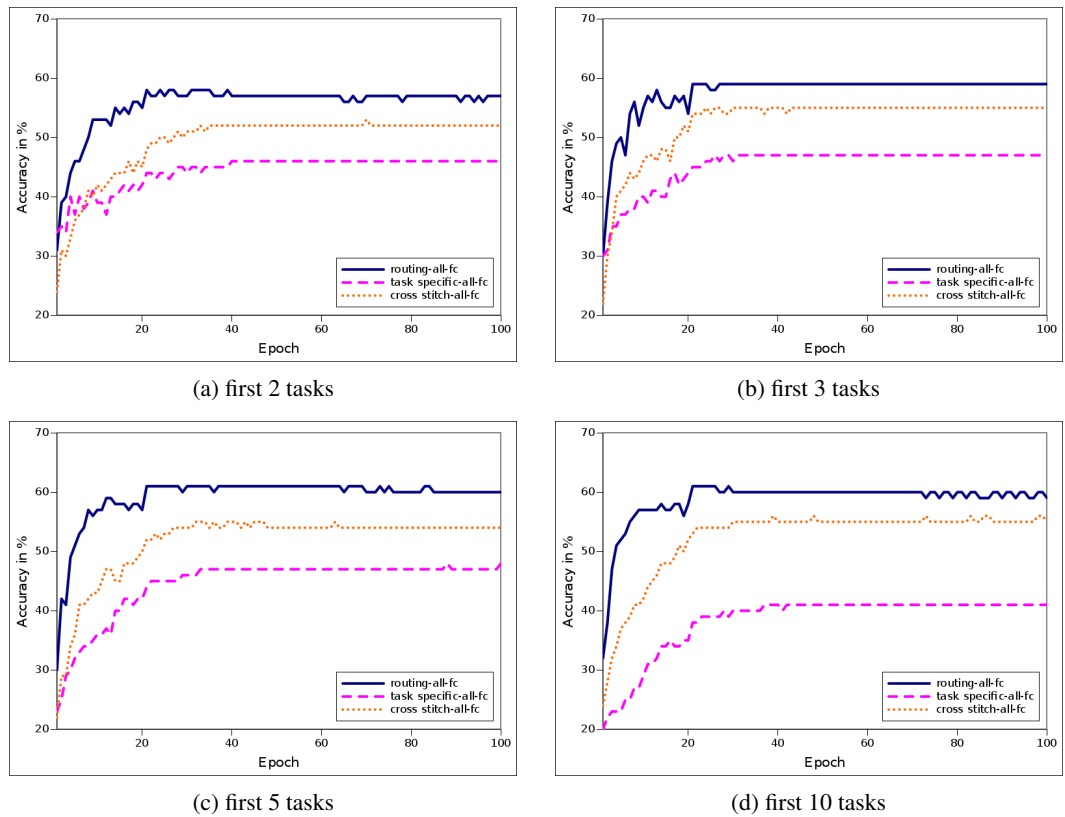

(a) first 2 tasks

(b) first 3 tasks

(c) first 5 tasks

(d) first 10 tasks

Figure 15: Results on the first $n$ tasks of CIFAR-MTL

| Name | Num Agents | Policy Representation | Part of State = (v, t, d) Used |
|---|---|---|---|
| MARL:WPL | Num Tasks | Tabular (num layers x num function blocks) | t, d |
| REINFORCE | Num Tasks | Vector (num layers) of approx functions | v, t, d |
| Q-Learning | Num Tasks | Vector (num layers) of approx functions | v, t, d |
| Q-Learning | Num Tasks | Tabular (num layers x num function blocks) | t, d |

Table 3: Implementation details for Figure 4. All approx functions are 2 layer MLPs with a hidden dim of 64.

| Name | Num Agents | Policy Representation | Part of State = (v, t, d) Used |
|---|---|---|---|
| routing-all-fc | Num Tasks | Tabular (num layers x num function blocks) | t, d |
| routing-all-fc non-layered | Num Tasks | tabular (num layers x num function blocks) | t, d |
| soft-routing-all-fc | Num Tasks | Vector (num layers) of appox functions | v, t, d |
| dispatched-routing-all-fc | Num Tasks + 1 | Vector (num layers) of appox functions + dispatcher | v, t, d |
| single-agent-routing-all-fc | 1 | Vector (num layers) of approx functions) | v, t, d |

Table 4: Implementation details for Figure 5. All approx functions are 2 layer MLP's with a hidden dim of 64.

**Policy Representation**   There are two dominant representation variations, as described in 3.1. In the first, the policy is stored as a table. Since the table needs to store values for each of the different layers of the routing network, it is of size num layers× num actions. In the second, it is represented either as vector of MLP's with a hidden layer of dimension 64, one for each layer of the routing network. In this case the input to the MLP is the representation vector $v$ concatenated with a one-hot representation of the task identifier.

**Policy Input**   describes which parts of the state are used in the decision of the routing action. For tabular policies, the task is used to index the agent responsible for handling that task. Each agent then uses the depth as a row index into into the table. For approximation-based policies, there are two variations. For the single agent case the depth is used to index an approximation function which takes as input concat($v$, one-hot($t$)). For the multi-agent (non-dispatched) case the task label is used to index the agent and then the depth is used to index the corresponding approximation function for that depth, which is given concat($v$, one-hot($t$)) as input. In the dispatched case, the dispatcher is given concat($v$, one-hot($t$)) and predicts an agent index. That agent uses the depth to find the approximation function for that depth which is then given concat($v$, one-hot($t$)) as input.

### 7.4   EXPLANATION OF THE WEIGHTED POLICY LEARNER (WPL) ALGORITHM

The WPL algorithm is a multi-agent policy gradient algorithm designed to help dampen policy oscillation and encourage convergence. It does this by slowly scaling down the learning rate for an agent after a gradient change in that agents policy. It determines when there has been a gradient change by using the difference between the immediate reward and historical average reward for the action taken. Depending on the sign of the gradient the algorithm is in one of two scenarios. If the gradient is positive then it is scaled by $1 - \pi(a_i)$. Over time if the gradient remains positive it will cause $\pi(a_i)$ to increase and so $1 - \pi(a_i)$ will go to 0, slowing the learning. If the gradient is negative then it is scaled by $\pi(a_i)$. Here again if the gradient remains negative over time it will cause $\pi(a_i)$ to decrease eventually to 0, slowing the learning again. Slowing the learning after gradient changes dampens the policy oscillation and helps drive the policies towards convergence.

