# OpenReview forum: "Routing Networks: Adaptive Selection of Non-Linear Functions for Multi-Task Learning"
_ICLR.cc/2018/Conference — Accept (Poster)_

### Official Review · AnonReviewer3 · 2017-11-26
**The paper is not clear enough.**

**Rating:** 6
**Confidence:** 3

**Review:**

Summary:
The paper suggests to use a modular network with a controller which makes decisions, at each time step, regarding the next nodule to apply. This network is suggested a tool for solving multi-task scenarios, where certain modules may be shared and others may be trained independently for each task. It is proposed to learn the modules with standard back propagation and the controller with reinforcement learning techniques, mostly tabular.

-	page 4:
In algorithm 2, line 6, I do not understand the reward computation. It seems that either a _{k+1} subscript index is missing for the right hand side R, or an exponent of n-k is missing on \gamma. In the current formula, the final reward affects all decisions without a decay based on the distance between action and reward gain. This issue should be corrected or explicitly stated.

The ‘collaboration reward’ is not clearly justified: If I understand correctly, It is stated that actions which were chosen often in the past get higher reward when chosen again. This may create a ‘winner takes all’ effect, but it is not clear why this is beneficial for good routing. Specifically, this term is optimized when a single action is always chosen with high probability – but such a single winner does not seem to be the behavior we want to encourage.

-	Page 5: It is not described clearly (and better: defined formally) what exactly is the state representation. It is said to include the current network output (which is a vector in R^d), the task label and the depth, but it is not stated how this information is condensed into a single integer index for the tabular methods. If I understand correctly, the state representation used in the tabular algorithms includes only the current depth. If this is true, this constitutes a highly restricted controller, making decisions only based on depth without considering the current output.
-	The functional approximation versions are even less clear: Again it is not clear what information is contained in the state and how it is represented. In addition it is not clear in this case what network architecture is used for computation of the policy (PG) or valkue (Q-learning), and how exactly they are optimized.
-	The WPL algorithm is not clear to me
o	 In algorithm box 3, what is R_k? I do not see it defined anywhere. Is it related to \hat{R}? how?
o	Is it assumed that the actions are binary?
o	I do not understand why positive gradients are multiplied with the action probability and negative gradients with 1 minus this probability. What is the source of a-symmetry between positive and negative gradients?

-	Page 6:
o	It is not clear why MNist is tested over 200 examples, where there is a much larger test set available
o	In MIN-MTL I do not understand the motivation from creating superclasses composed of 5 random classes each: why do we need such arbitrary and un-natural class definitions?

-	Page 7:
The results on Cifar-100 are compared to several baselines, but not to the standard non-MTL solution: Solve the multi-class classification problem using a softmax loss and a unified, non routing architecture in which all the layers are shared by all classes, with the only distinction in the last classification layer. If the routing solution does not beat this standard baseline, there is no justification for its more complex structure and optimization.

-	Page 8: The author report that when training the controller with single agent methods the policy collapses into choosing a single module for most tasks. However, this is not-surprising, given that the action-based reward (whos strength is unclear) seems to promote such winner-takes-all behavior.

Overall:
-	The paper is highly unclear in its method representation
o	There is no unified clear notation. The essential symbols (states, actions, rewards) are not formally defined, and often it is not clear even if they are integers, scalars, or vectors. In notation existing, there are occasional notation errors.
o	The reward is a) not clear, and b) not well motivated when it is explained, and c) not explicitly stated anywhere: it is said that the action-specific reward may be up to 10 times larger than the final reward, but the actual tradeoff parameter between them is not stated. Note that this parameter is important, as using a 10-times larger action-related reward means that the classification-related reward becomes insignificant.
o	The state representation used is not clear, and if I understand correctly, it includes only the current depth. This is a severely limited state representation, which does not enable to learn actions based on intermediate results
o	The continuous versions of the RL algorithms are not explained at all: no state representation, nor optimization is described.
o	The presentation suffers from severe over-generalization and lack of clarity, which disabled my ability to understand the network and algorithms for a specific case. Instead, I would recommend that in future versions of this document a single network, with a specific router  and set of decisions, and  with a single algorithm, will be explained with clear notation end-to-end

Beyond the clarity issues, I suspect also that the novelty is minor (if the state does not include any information about the current output) and that the empirical baseline is lacking. However, it is hard to judge these due to lack of clarity.


After revision:
- Most of the clarity issues were handled well, and the paper now read nicely
- It is now clear that routing is not done based on the current input (an example is not dynamically routed based on its current representation). Instead routing depends on the task and depth only.  This is still interesting, but is far from reaching context-dependent routing.
- The results presented are nice and show that task-dependent routing may be better than plain baseline or the stiching alternative.  However, since this is a task transfer issue, I believe several data size points should be tested. For example, as data size rises, the task-specific-all-fc alternative is expected to get stronger (as with more data, related task are less required for good performance).


-

---

> ### Author Response · Authors · 2017-12-22
> **Response to your review**
>
> Thanks for pointing out many places where we were less than clear.  We have re-written or edited many sections of the paper (mostly the earlier sections) to address both the specific typos you point out as well as more systemic issues in clarity.  In particular, we have taken your advice and made the presentation less general, focusing on a clear description of what we implemented and employing a consistent notation.  We think this has improved the paper.
>
> Responses to questions and comments below:
>
> -The ‘collaboration reward’ is not clearly justified: If I understand correctly, It is stated that actions which were chosen often in the past get higher reward when chosen again. This may create a ‘winner takes all’ effect, but it is not clear why this is beneficial for good routing. Specifically, this term is optimized when a single action is always chosen with high probability – but such a single winner does not seem to be the behavior we want to encourage.
>
> As we have clarified in the new version, the collaboration reward was intended to test the ability of the network to learn a more compressed routing scheme while not sacrificing performance.  Because we don't know ahead of time how many function blocks per layer are really required to solve the problem (we use #function blocks = #tasks) we wanted to see if we could encourage the network to use fewer.  We have added a comparison of the effect of the collaboration reward using the hyper-parameter multiplier rho which ranges from 0 (no collaboration reward) to 10 (large collaboration reward).  Figure 12 in the appendix shows that to a large extent the performance is robust to changes in rho in the [0, 1] range on CIFAR-MTL using the WPL algorithm. There is the potential for this reward to produce a lack of diversity in the function block choices but this doesn’t happen empirically in the multi-agent approaches.  And we see a lack of diversity in the choices made by the single agent approach even with no collaboration reward (rho = 0.0).
>
> -Page 5: It is not described clearly (and better: defined formally) what exactly is the state representation. It is said to include the current network output (which is a vector in R^d), the task label and the depth, but it is not stated how this information is condensed into a single integer index for the tabular methods. If I understand correctly, the state representation used in the tabular algorithms includes only the current depth. If this is true, this constitutes a highly restricted controller, making decisions only based on depth without considering the current output.
>
> We hope we’ve made this clearer in the paper.  First, a state is as you say a triple of (v, t, i) where v is in R^d, t is an integer task label, and i is the current recursion depth.  Let us take the WPL-router as an example of how this works for a tabular approach.  WPL has one agent per task.  Each agent has a table of size max depth x num function blocks which holds the action probabilities for each layer.  So the routing function depends on both the task label and depth not just the depth.  The individual policy decision function of each task-agent uses only the depth to decide its route.  We have also experimented with a "dispatched" approach which adds an extra dispatching agent which learns pick the agent used to route the instance rather than using the task label to index it.  This approach (called "routing-all-fc dispatched" in Figure 5) uses v and t to pick the router agent, which itself uses depth to make its decision.  We have added a table to the appendix which states explicitly which parts of the state are used for each approach used in Figures 4 and 5.
>
> - The functional approximation versions are even less clear: Again it is not clear what information is contained in the state and how it is represented. In addition it is not clear in this case what network architecture is used for computation of the policy (PG) or valkue (Q-learning), and how exactly they are optimized.
>
> We have clarified this in the paper.  The network architecture used for approximation is a 2 layer MLP with hidden dimension of size 64.  The state is encoded by converting the task and depth components to one-hot vectors and concatenating them to the input vector representation (if used).  All training is performed with SGD (hyperparameters described in the text).  We also tried with Adam but it performed marginally less well.
>
> -The WPL algorithm is not clear to me
>
> We have added a more detailed explanation to the appendix in Section 7.6.
>
> -In algorithm box 3, what is R_k? I do not see it defined anywhere. Is it related to \hat{R}? how?
>
> The algorithms have been completely reworked to make clear their inputs and outputs.  R_i is the return for action a_i (the sum of the future rewards + a final reward).  \hat{R}_i is the running average return for action a_i over the samples in the dataset.
>
>
>
> continued in next reply

---

> > ### Author Response · Authors · 2017-12-22
> > **Response to review (cont.)**
> >
> > -Is it assumed that the actions are binary?
> >
> > Actions are in the set {1, ..., k, PASS} where k is the number of function blocks and PASS is a special action which just causes the iteration to be skipped (the state remains the same).
> >
> > -I do not understand why positive gradients are multiplied with the action probability and negative gradients with 1 minus this probability. What is the source of a-symmetry between positive and negative gradients?
> >
> > Repeated from Section 7.6: The WPL algorithm is a multi-agent policy gradient algorithm designed to help dampen policy oscillation and encourage convergence.  It does this by slowly scaling down the learning rate for an agent after a gradient change in that agent’s policy.  It determines when there has been a gradient change by using the difference between the immediate reward and historical average reward for the action taken.  Depending on the sign of the gradient the algorithm is in one of two scenarios.  If the gradient is positive then it is scaled by 1-pi(a_i).  Over time if the gradient remains positive it will cause pi(a_i) to increase and so 1-pi(a_i) will go to 0, slowing the learning.  If the gradient is negative then it is scaled by pi(a_i).  Here again if the gradient remains negative over time it will cause pi(a_i) to decrease eventually to 0, slowing the learning again. Each such slowing helps dampen the variations or oscillations in the policies and eventually to help them converge.
> >
> > - Page 6:
> > - It is not clear why MNist is tested over 200 examples, where there is a much larger test set available
> > MNIST is actually tested on 200 samples *per task*, which for 10 tasks is 2,000 examples in total. We can increase the number of test samples and re-run the experiments if you feel that this is insufficient.
> >
> > - In MIN-MTL I do not understand the motivation from creating superclasses composed of 5 random classes each: why do we need such arbitrary and un-natural class definitions?
> >
> > First, as a practical matter, mini-ImageNet does not have superclasses to use as a natural tasks (e.g. Fish -> {goldfish, flounder, carp}) the way that CIFAR-100 does.  Second, in real-world problems what constitutes a "task" may be quite far from what we would consider "natural" taxonomic classification.  Imagine, for example, we collect data every hour from a website for several weeks.  We can treat each hour in the day as a separate task.  Here there will likely be some intra-task similarities and some inter-task differences but not in as coherent or natural a way as in image classification on CIFAR.  And we would like to understand if MTL will be helpful in this regime as well.  A task with a randomly chosen label set tests our ability to handle tasks with less coherence.  It's a logical extreme in which we might expect relatively less positive transfer and potentially more negative transfer, so worth examining.
> >
> > -Page 7:
> > The results on Cifar-100 are compared to several baselines, but not to the standard non-MTL solution: Solve the multi-class classification problem using a softmax loss and a unified, non routing architecture in which all the layers are shared by all classes, with the only distinction in the last classification layer. If the routing solution does not beat this standard baseline, there is no justification for its more complex structure and optimization.
> >
> > We performed this experiment early on many times, but since the results were so much worse than either of the baselines we chose, omitted it from the final results.  We will run it again with the current hyper-parameters and post the results before the end of the review period.
> >
> >  -Page 8: The author report that when training the controller with single agent methods the policy collapses into choosing a single module for most tasks. However, this is not-surprising, given that the action-based reward (whos strength is unclear) seems to promote such winner-takes-all behavior.
> >
> > We have tried this with no collaboration reward (rho=0.0) on and see exactly the same behavior.  We believe that it is more a function of a shared policy for all tasks than any particular reward structure.

---

> ### Author Response · Authors · 2018-01-04
> **Results of fully shared model on CIFAR**
>
> As requested, we ran the experiment with a single fully shared model (no task-specific fc layers) on CIFAR.  It achieves 37% accuracy (averaged over 2 runs) after 100 epochs of training.  This is below the lowest baseline (task-specific-all-fc) which achieves 42%.

---

> ### Author Response · Authors · 2018-01-04
> **Per-task data size experiments**
>
> Thanks for the suggestion.  We will include a chart on this in the final draft.

---

### Official Review · AnonReviewer2 · 2017-11-27
**Good work**

**Rating:** 8
**Confidence:** 4

**Review:**

In this paper, the authors present a novel formulation for learning the optimal architecture of a neural network in a multi-task learning framework. Using multi-agent reinforcement learning to find a policy — a mapping from the input layer (and task indicator) to the function block that must be used at the next layer, the paper shows improvement over hard-coded architectures with shared layers.

The idea is very interesting and the paper is well-written. The extensive review of the existing literature, along with systematically presented qualitative and quantitative results make for a clear and communicative read. I do think that further scrutiny and research would benefit the work since there are instances when the presented approach enjoys some benefits when solving the problems considered. For example, the more general formulation that uses the dispatcher does not appear to work ver well (Fig. 5), and the situation is only improved by using a task specific router.

Overall, I think the idea is worthy of wide dissemination and has great potential for furthering the applications of multi-task learning.

---

> ### Author Response · Authors · 2017-12-22
> **Response to review**
>
> -The idea is very interesting and the paper is well-written. The extensive review of the existing literature, along with systematically presented qualitative and quantitative results make for a clear and communicative read.
>
> Thanks!
>
> -I do think that further scrutiny and research would benefit the work since there are instances when the presented approach enjoys some benefits when solving the problems considered. For example, the more general formulation that uses the dispatcher does not appear to work ver well (Fig. 5), and the situation is only improved by using a task specific router.
>
> We agree.  The dispatched router which tries to learn which agent to send the instance to doesn't perform as well.  It would be even more beneficial (but harder still) to do dispatched routing without using the task label at all.  This would greatly expand the applicability of the approach (it could then be used potentially to improve any neural network on any dataset).  We are actively investigating ways to improve this...

---

### Official Review · AnonReviewer1 · 2017-11-27
**Revised Review**

**Rating:** 7
**Confidence:** 3

**Review:**

The paper introduces a routing network for multi-task learning. The routing network consists of a router and a set of function blocks. Router makes a routing decision by either passing the input to a function block or back to the router. This network paradigm is tested on multi-task settings of MNIST, mini-imagenet and CIFAR-100 datasets.

The paper is well-organized and the goal of the paper is valuable. However, I am not very clear about how this paper improves the previous work on multi-task learning by reading the Related Work and Results sections.

The Related Work section includes many recent work, however, the comparison of this work and previous work is not clear. For example:
"Routing Networks share a common goal with techniques for automated selective transfer learning
using attention (Rajendran et al., 2017) and learning gating mechanisms between representations
(Stollenga et al., 2014), (Misra et al., 2016), (Ruder et al., 2017).  However, these techniques have
not been shown to scale to large numbers of routing decisions and task." Why couldn't these techniques scale to large numbers of routing decisions and task? How could the proposed network in this paper scale?

The result section also has no comparison with the previously published work. Is it possible to set similar experiments with the previously published material on this topic and compare the results?


-- REVISED

Thank you for adding the comparisons with other work and re-writing of the paper for clarity.
I increase my rating to 7.

---

> ### Author Response · Authors · 2017-12-22
> **Response to your review**
>
> The Related Work section includes many recent work, however, the comparison of this work and previous work is not clear. For example: "Routing Networks share a common goal with techniques for automated selective transfer learning using attention (Rajendran et al., 2017) and learning gating mechanisms between representations (Stollenga et al., 2014), (Misra et al., 2016), (Ruder et al., 2017).  However, these techniques have not been shown to scale to large numbers of routing decisions and task."
>
> -Why couldn't these techniques scale to large numbers of routing decisions and task? How could the proposed network in this paper scale?
>
> The papers by Misra and Ruder experiment with only 2 tasks at a time so it was not clear if they would scale to more tasks.  These approaches make 2 copies of a convnet and connect them at a small number of junctures (e.g. 5) with cross-stitch or sluice connections to allow inter-task sharing.  This is a soft-attention approach which computes the input for each function block at layer i as a linear combination of the activations of the function blocks at layer i-1.  If there are k such function blocks at each layer, this introduces O(k^2) additional parameters for each connection.  By comparison for routing using the WPL (tabular) approach we need no additional parameters and only O(num tasks x max depth x num function blocks) additional memory.
>
> We have now re-implemented Misra and applied it to 2, 3, 5, 10, and 20 task problems for CIFAR-100.  In all cases routing networks are uniformly better (on a separate test set) over the entire course of training by a large margin.  We have conducted scaling experiments (see the Appendix) which show that the per-task training cost is effectively constant for routing networks but grows linearly for cross-stitch networks.  In addition, routing networks are significantly faster for all numbers of tasks, achieving per-task time improvements of approximately 2-6X on the 2, 3, 5, and 10 task experiments.  These per-task improvements translate to real gains in training time.  For example, on CIFAR-100 (20 task), we see an average (over 3 runs) training time improvement of 85% (5.6 hours for routing nets to reach cross-stitch final performance levels achieved after 38 hours of training).
>
> -The result section also has no comparison with the previously published work. Is it possible to set similar experiments with the previously published material on this topic and compare the results?
>
> Yes, see the new comparison with Misra's cross-stitch networks.

---

### Public Comment · ~Chrisantha_Fernando1 · 2017-12-18
**Learned routing vs. Evolved routing.**

Having experimented with learned routing in order to extend our pathNet work, I was very interested in the authors insight that an RL algorithm which explicitly rewards convergence to the same pathway was apparently critical. This makes sense. Previous methods of gating such as "Outrageously large nets" have the opposite diversity cost, but in RL where it is necessary to train a pathway hard for some time, the opposite prior is clearly beneficial. I think this is an important insight of the authors. In  pathNet this was not required as evolutionary dynamics automatically achieved this convergence to a single pathway per task.

Also, much like our pathNet work the authors found it was important to reset the RL agent for each task, just as we reset the population of pathways at each task. We also found such a resetting to be critical. Finally, whereas we focus on transfer learning the authors focus on multi-task learning which has the additional constraint that it is not possible to fix modules learned in previous tasks. Also, I was interested in the output of the previous module being important for the router. I would like to see a test of whether this is critical, or whether it is enough to see the observation and task label. Another difference is that gating is not combinatorial in this paper, only using one module per layer instead of K per layer.

PathNet: https://arxiv.org/abs/1701.08734

Many thanks for a fascinating paper.

---

> ### Author Response · Authors · 2018-01-02
> **Thoughts on routing and pathnets**
>
> Hi Chrisantha -- thanks for the nice comment and making this interesting connection to your work on pathnets!
>
> We think there is a clear overlap in the way we are both thinking about this. In particular having a task-specific "agent" (whether in the sense of the evolutionary approach or the RL approach) seems to give benefit over the shared (single agent) approach.  We too noted during our experiments that diversity was not an issue for the multi-agent case (though it was for the single agent) and originally introduced the collaboration reward to see if we could encourage the multi-agent approach to use fewer paths without sacrificing performance.  The hyper-parameter rho is a knob that allows us to experiment with this.  Since we put up the first version of the paper we have conducted additional experiments on rho (see the new version Appendix, Sec. 7.1) which show that it is marginally helpful for this task.  We found that it wasn't helpful to use rho for the CIFAR and MIN experiments so there set it to 0, but it was useful for MNIST and there we set it to 0.3.
>
> On the point about using the previous modules activation (output) for the router decision-making we've added a table to the appendix (Table 3) which shows which of the RL algorithms use which parts of the state.  The state for us consists of three values (v, t, d), where v is the previous layer's activation (or the input), t is an integer task label, and d is the current recursion depth.  The most successful approach for us was the WPL (multi-agent) trainer which uses just t and d. That approach is tabular which means that it is not possible to add v to the input directly (though we are looking at approximator versions that could do it...).  We did try a few single agent RL algorithm variations which do use v -- for example the PG and Q (approximator) versions -- but these didn't do as well.  Appendix table 5 also  compares against the multi-agent case (dispatched-routing-all-fc) and the single agent case (single-agent-routing-all-fc) but these also were not able to match the tabular WPL.  We think there is clearly potential for approaches which use the previous activation to do better than WPL but doing so greatly increases the size of the search space and we think this does more harm than good in the current training approach.  We're working on it though...

---

### Author Response · Authors · 2017-12-22
**Response to Reviewers**

We want to thank all the reviewers for their careful reading and thoughtful comments.  We have tried to address all the issues and questions raised in a new revision of the paper.

TL;DR:
1. Comparison to Cross-stitch networks (CVPR 2016, oral) for multi-task learning over which we show significant gains in accuracy for 2, 3, 5, 10, and 20 tasks; a constant vs. linear time scaling of per-function-block training cost and a roughly 85% average reduction in actual training time for routing nets to achieve cross-stitch accuracy levels (38 hours for cross-stitch vs 5.6 hours for routing nets on CIFAR-MTL)

2. Extensive re-writing of the paper for clarity and consistency.

3. Additional experiments showing the performance effect of the collaboration reward rho as well as comparison to cross-stitch and the baselines for 2, 3, 5, and 10 task problems.

Specific responses in separate replies to each reviewer.

---

### Decision · Program_Chairs · 2018-01-29
**ICLR 2018 Conference Acceptance Decision**

**Decision:**

Accept (Poster)

**Comment:**

The proposed routing networks using RL to automatically learn the optimal network architecture is very interesting. Solid experimental justification and comparisons. The authors also addressed reviewers' concerns on presentation clarity in revisions.